# How Do Different Households Respond to Public Education Spending?

**Shuang Yu** * **and Xiaojun Zhao**

School of Economics, Peking University, Beijing 100871, China; zhaoxiaojun@pku.edu.cn
* Correspondence: yushuang1995@pku.edu.cn; Tel.: +86-188-1168-7987

**Abstract:** Using data from the China Family Panel Studies (CFPS), we developed an educational production function to examine how households with different income levels and parental human capital respond to changes in public spending. Our results suggest that there is a significant complementary effect between household inputs of time and money and public investments in the educational process. However, the results are heterogeneous in terms of different income levels. Rich families have more incentives to invest in their children, suggesting a crowd-in effect of public resources. In contrast, public spending crowds out private inputs for poor families, who care more about their own well-being. Moreover, we show that educational investments in parents have spill-over effects on their children, but the degrees of influence are different for the poor and the rich.

**Keywords:** public education spending; household inputs; income level; human capital

## 1. Introduction

Many economists have paid much attention to the economic growth theory since Adam Smith explained the nature and factors of national wealth. In the 1950s and 1960s, Gary Becker, T. W. Schultz, and others began to focus on the roles education and human capital play in the economy. Recently, economists have combined these two theories, examining the relationship between long-term economic growth and education [1–3].

At the macro level, several studies have found that public education spending can reduce poverty, diminish the income gap, and boost growth [4,5]. However, we should also consider the micro-mechanism of education, which is a social practice activity that affects people's physical and mental development. Previous studies have found positive effects of school financial resources on students' performance [3,6,7], while few studies have suggested negligible or even negative effects [8].

Our research shows that two factors may account for mutually exclusive results. The first essential factor that has been overlooked so far in the literature is that the family environment has huge leverage over children's minds. In addition to the government, parents play a role in educating their children [9]. Parents can choose the methods and type of education their children receive, and in general, parental teaching styles, habits, and daily behaviors have a large influence on their children. Children who live in families with higher-educated parents form good habits, have better chances of receiving a better education, and have higher grades in school.

Another important factor is the household input—consisting of time and money—and the role it plays in the children's performance. Families respond to public spending education differently. There is a probable crowd-in effect for high-income and higher-educated households facing increased public education spending. They may spend more time accompanying their children and more money on the children's education, for example, helping them access smart education equipment and enrolling them in better schools and many tutoring classes. However, for the low-income and lower-educated households, government investment may crowd out the household input. As the government meets the



basic educational needs of their children, they are more likely to spend money on personal consumption instead of their children's education. Therefore, when studying the impact of public education investment on students' performance, we need to further consider family socioeconomic factors and household behavior towards public education spending.

This article mainly addresses the following questions: Is there a crowd-in or crowd-out effect between public education expenditure and household inputs in Chinese society? What are the roles of the income level and educational background in shaping the impact of government spending on individual behaviors? These questions need to be answered theoretically and empirically. In this study, we began with an educational production function that includes household spending and parental effort as inputs for children's human capital, examining the impact of household income on preferences for public education and other economic factors. Then, we used household survey data and subdivided provincial education expenditure data as the analysis objects, studying how government education spending affects private education expenditures and parental effort from a micro perspective. By examining families' responses to various types of household expenses, we evaluated the improvement of teaching quality by government education investments.

Most recent studies regard the private inputs and public education expenditures as either substitutes or complements. For instance, in some studies, it was found that increases in public education resources replaced private inputs in both time and money. Das et al. [10] showed that households offset their educational spending in response to anticipated school inputs, leading to decreased test scores. Gamlath and Lahiri [11] developed an overlapping generation model to show that there is strong substitutability between public education and private expenditures. They also mentioned that it has better economic outcomes because families could spend more on consumption and investment. Kim [12] showed that with an increase in school expenditure, lower-educated mothers decreased their child care time but higher-educated mothers did not change their behaviors. Houtenville and Conway [13] shared the same opinion that parents reduce their effort in response to improved school resources, suggesting a "crowding out" effect of school investments. Liu, Mroz, and van der Klaauw [14] implied that families choose where to live or work based on school quality.

On the other hand, one author suggested that governments only provide a baseline level of education and health care, and individuals could pursue a higher level of living quality [15]. Meanwhile, Blankenau et al. [16] and Arcalean and Schipou [17] suggested that basic education financed by the government is only a prerequisite for receiving tertiary education. Our results should, therefore, be interpreted as evidence that income status and education level shape families' behaviors with increases in government resources.

The rest of this paper is organized as follows. Section 2 presents the theoretical framework of private and public education. Section 3 describes the summary statistics of the main variables. Section 4 provides the estimation results. Section 5 concludes the paper. Several proofs and computations are supplied in Appendix A.

## 2. Model

### 2.1. Conceptual Framework

In the model, each family is composed of an altruistic parent and a child. The model consists of two periods. In the first period, the parent is middle-aged and the child is young. In the second period, the parent is retired and the child is middle-aged. The parent's utility function is then expressed as the following:

$$U_P = u(c_1) + u(c_2) + \rho U_k(c_k) \tag{1}$$

where $\rho > 0$ reflects the degree of parent altruism, and subscripts p and k represent the parent and child, respectively. The total utility of the parent, $U_P$, includes the two-stage utility of consumption in two periods, $u(c_i)$, i = 1,2, and the child's utility of consumption when middle-aged, $U_k(c_k)$. Both $u$ and $U_k$ satisfy the strictly concave, twice continuous differentiability, and Inada conditions. For simplicity, we assume that the time discount factor is 1.

The parent's budget constraints are represented by the following:

$$c_1 = w_p h_p (1 - z) - s - q \tag{2}$$

$$c_2 = s(1 + r_s) - b \tag{3}$$

In the first period, $h_p$ is the parent's human capital level, which varies across families, and they receive the after-tax wage, $w_p$. $z \in (0, 1)$, indicates the proportion of time parents need to invest in raising each child and, therefore, $1 - z$ is the parents' working hours. Income is mainly used for consumption, $c_1$, savings, s, and investments in the child's human capital, q. In the second period, the parent's after-tax rate of return to savings, $s(1 + r_s)$, is used for their own consumption, $c_2$, and a bequest, b, to a child.

The child's consumption equals his/her labor income and possibly a bequest, b, from the parents:

$$c_k = w_k h_k + b \tag{4}$$

where $h_k$ is the child's human capital level, and $w_k$ is the after-tax wage rate of the child. The child's human capital level, $h_k$, is produced by the parent's human capital level, $h_p$, the time parent spends on the child, z, the public education spending, g, and the private education spending, q:

$$h_k = f(h_p, z, q, g) \tag{5}$$

which is assumed to be strictly concave. The partial derivatives, $f_i > 0$, $f_{ii} < 0$, ($i = h_p, z, q, g$), and the Inada conditions are fulfilled for all inputs, $\lim_{i \to 0} f_i = +\infty$, $\lim_{i \to +\infty} f_i = 0$, which ensures that q and z are positive. We assume that the level of public education investment, g, is the same for all children in the economy, so a higher level of public education spending will increase everyone's human capital ceteris paribus. The influence of parents on children's human capital is divided into three aspects: parental human capital, $h_p$, determines their parenting style, which will affect the child's cognitive ability, behavior, and personality; parent's educational investments, q, have an influence on the child's growth environment; the time, z, that parents put into their child's education will affect the child's personality and ways of thinking. These three channels of influence interact with each other. High-income and better-educated parents spend more time and money on their children's education than disadvantaged parents do, so children from families with high socioeconomic status can receive better care and education, and thus, have better future development (by enjoying additional advantages, such as electronic devices, books, and summer camp).

We obtained the parent's maximization problem as follows:

$$\max_{\{q,b,z,s\}} U_p = u_1(w_p h_p (1-z) - s - q) + u_2(s(1+r_s) - b) + \rho U_k(w_k h_k + b) \tag{6}$$

$$\text{s.t. } q \geq 0, b \geq 0 \tag{7}$$

The first-order conditions with respect to q, b, z, and s are as follows:

$$q: \frac{\partial U_p}{\partial q} = -\acute{u_1} + \rho \acute{U_k} w_k f_q = 0 \tag{8}$$

$$b: \frac{\partial U_p}{\partial b} = -\acute{u_2} + \rho \acute{U_k} = 0 \tag{9}$$

$$z: \frac{\partial U_p}{\partial z} = -\acute{u_1} w_p h_p + \rho \acute{U_k} w_k f_z = 0 \tag{10}$$

$$s: \frac{\partial U_p}{\partial s} = -\acute{u_1} + \acute{u_2}(1 + r_s) = 0 \tag{11}$$

Combining (7)–(9) then gives the following arbitrage-like condition:

$$w_k f_q = 1 + r \tag{12}$$

$$w_p h_p f_q = f_z \tag{13}$$

which implies that, at the optimum, the marginal benefit of investing one dollar in a child's human capital should be equal to the constant rate of return to saving that money; see $q^*$ in Figure 1. Meanwhile, the marginal benefit of one dollar in the child's human capital is equal to the marginal benefit of the child's human capital in a proportion of the investment; see $q^*$ in Figure 2.

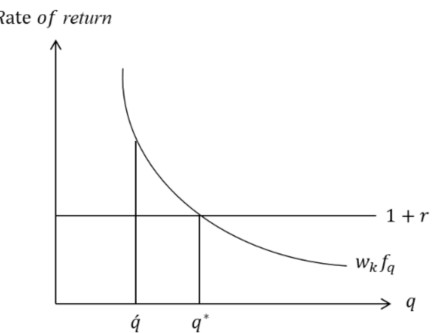

**Figure 1.** Return of human capital investments from the parent's viewpoint.

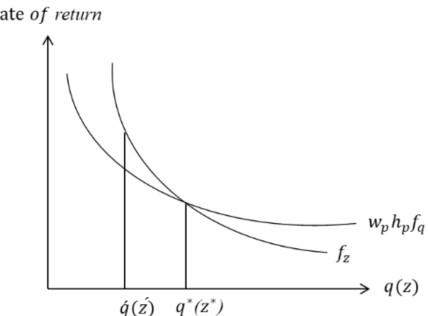

**Figure 2.** Balance between time and money for human capital from the parent's perspective.

### 2.2. Policy

Let us consider an economy consisting of one rich family and one poor family living in the country and in the city, respectively. If the main goal for the government is to achieve equity between the rich and the poor, one important task is to ensure the wellbeing of the poor people. Thus, it is critical to determine which kinds of policies can narrow the gap between the rich and the poor. When the government increases public education expenditure, it covers part of the children's tuition, miscellaneous fees, and living expenses, reducing the amount of funds originally used by the children's families for education spending. At this time, families face a choice of how to use this spare money. The marginal rate of return on consumption is high for poor families, so they probably spend more on consumption but not on children's education. In contrast, rich families face a development but not an existential problem, and they spend more on their children. This understanding can facilitate the analysis of the effect of government education policies and family behaviors.

Our central hypothesis is that parents may respond differently, in terms of spending and time, to public spending and other factors. The rich parent spends more time and money on their child's education with increased government spending, showing a complementarity effect. The poor parent, who cares more about their survival, will choose to spend on consumption but not on their child's development. These results may provide some new ideas for traditional policymakers.

All the results are connected with the parent's wage, $w_p$, and the child's wage, $w_k$, so we used a comparative static analysis method to study the effect. In addition, we concentrated on the influence of parental income and education on their decision-making,

so we transformed the following formulas as a function of parent wages, $w_p$, based on the different categories of children's wages, $w_k$.

**Lemma 1.** *Given other conditions, $b = 0$ is one of the optimal solutions. An equivalent adjustment mechanism can be formed. The result holds when the second period of the bequest, $b$, is used for the first period of private education spending $q$.*

**Proof.** b $= 0$ and b $> 0$ are two optimal solutions for this question. To simplify the computation, we can choose b $= 0$ to replace other solutions. When $r_s$ is constant, s decreases b$/(1 + r_s)$, and $c_1$, $c_2$, and $c_k$ do not change. The total utility remains the same, which is still the optimal solution. So we can set bequest, b, to 0, only considering the change in private education spending, q.

2.2.1. Public Education

Social planners use public education as an effective way to narrow the gap between the poor and the rich. However, in real life, parents' attitudes towards further investments into public education are strongly affected by their financial situation. If we consider their various income levels, not surprisingly, increasing public education spending aggravates income inequality. Nordblom [18] similarly suggested that high-income families take more advantage of increased productivity; thus, the educational gap between rich and poor children is not narrowed as expected. Kim [12] showed that less-educated parents significantly decrease children's care time when school expenditures increase, whereas highly-educated parents instead increase their time investment. However, Li [19] concluded that public education has a restrictive effect in the short term, while its long-term effect on poor families is beneficial for the next generation to improve their economic status.

From Equation (A3) in Appendix A, we can obtain the function for the child's expected wage, $w_k$, in Figure 3. Based on different levels of children's wages, there is a heterogeneous response to public spending between poor and rich families.

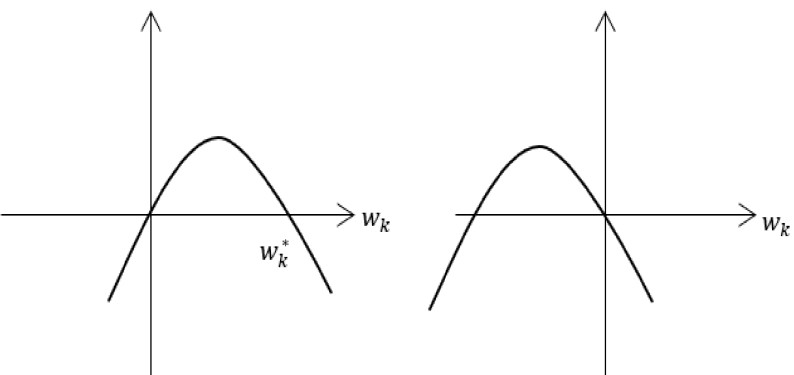

**Figure 3.** Function of child's wage rate.

The total effect on public education is:

$$\frac{\partial q}{\partial g} = \frac{-(u_1'' u_2'' w_p^2 h_p^2 (1+r)^2 + (\rho u_k'' w_k^2 f_z^2 + \rho u_k' w_k f_{zz})(u_1'' + u_2''(1+r)^2))(\rho u_k'' w_k^2 f_q f_g + \rho u_k' w_k f_{qg})}{|B_1|} +$$
$$\frac{-(u_1'' u_2'' w_p h_p (1+r)^2 + (\rho u_k'' w_k^2 f_z f_q + \rho u_k' w_k f_{zq})(u_1'' + u_2''(1+r)^2))(\rho u_k'' w_k^2 f_z f_g + \rho u_k' w_k f_{zg})}{|B_1|} \tag{14}$$

$$\frac{\partial z}{\partial g} = \frac{-(u_1'' u_2'' w_p h_p (1+r)^2 + (\rho u_k'' w_k^2 f_q f_z + \rho u_k' w_k f_{qz})(u_1'' + u_2'' (1+r)^2))(\rho \rho u_k'' w_k^2 f_q f_g + \rho u_k' w_k f_{qg})}{|B_1|} +$$
$$\frac{-(u_1'' u_2'' (1+r)^2 + (\rho \rho u_k'' w_k^2 f_q^2 + \rho u_k' w_k f_{qq})(u_1'' + u_2'' (1+r)^2))(\rho \rho u_k'' w_k^2 f_z f_g + \rho u_k' w_k f_{zg})}{|B_1|}$$ (15)

**Proposition 1.** There is an obvious substitution effect between private and public spending for the poor family. In contrast, the rich family adapts its investment strategies to enhance the child's expected income. In general, public education spending will expand the educational gap.

**Proof.** When a child's expected wage is less than $w_k^*$, the quadratic term, the coefficients of the linear and quadratic entries of $w_p$ are both positive. So, based on this, we can obtain Figure 4. According to this figure, if $w_p < w_p^*$, representing the poor family, $\partial q / \partial g < 0$, which means that if the government increases its educational investments, the family will decrease their educational spending. Hence, the total effect will be negative. For the rich family, when $w_p > w_p^*$, we could have $\partial q / \partial g > 0$, showing the positive effect between government educational spending and household spending. In this way, if parents believe their children will earn less in the future, the rich family will also take actions to improve spending on education when the government raises the financial costs of education. In contrast, the poor family chooses to decrease their educational spending as public education spending increases.

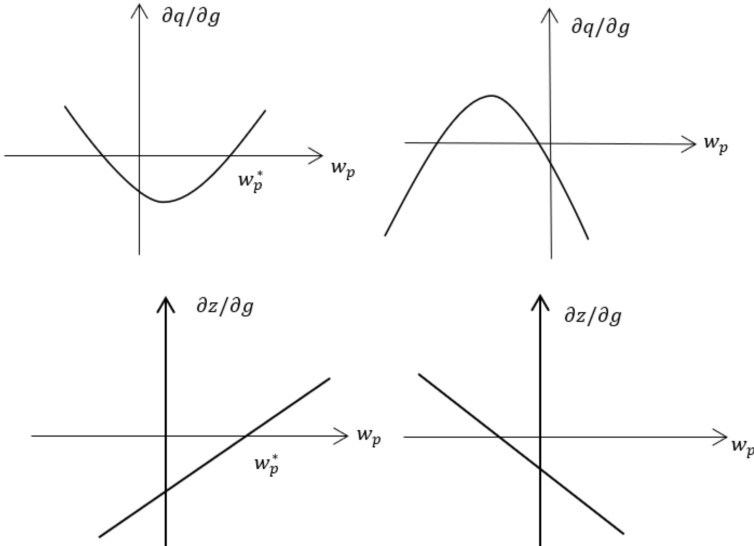

**Figure 4.** Function of household inputs to public education spending.

When a child's expected wage is more than $w_k^*$, the quadratic term, the coefficients of the linear and quadric entries of $w_p$ are both negative. So, based on this, we can obtain Figure 4. In this situation, private and public education spending show substitution effects for both the poor and the rich families. This reflects the families' beliefs that no matter what they do, their children are likely to have a bright future, so both families decrease investments in their children's education.

The relationship between parental time and public spending presents the same trend as the relationship between private and public spending. In the first condition, in which the children have low expected wages, there is a substitution effect between public education spending and parental effort for the poor family, while the effect becomes complementary for the rich family. In the second condition, when the children's expected wage is high, both families reduce their efforts for their children's development. The comprehensive interpretation of the conclusions is identical to that of the private spending results.

### 2.2.2. Parental Human Capital

There is no doubt that it is useful to continue education and work training. Apparently, highly-educated parents have a better capacity to absorb knowledge than lower-educated parents. Even if they are positively affected by increased human capital investments, the rich gain more profits.

**Proposition 2.** Increasing the human capital of parents increases their educational investments for their children. Because of different effects on the poor and the rich parent, educational inequality might increase between their children.

**Proof.** As shown in Figure 5, there is a positive correlation between parental human capital and private educational time and money spending for both the rich and the poor $\left(\frac{\partial q/z}{\partial h_p}\right)_{poor/rich} > 0$. However, the slope of the curve is different, suggesting that the rich parent spends more on their children's education. Therefore, inequality increases.

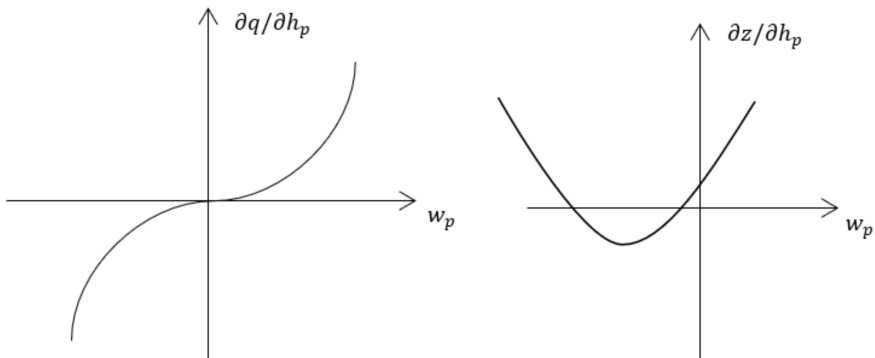

**Figure 5.** Function of household inputs to parental human capital.

### 2.2.3. Parental Wage Rate

It is generally believed that rising parental incomes, $w_p$, could promote social equality. However, when parents rather than children themselves choose how to use the income, it could well be the case that poor parents catch an opportunity to improve their well-being, and instead decrease the educational spending, while families in the top tier of income distribution have a strong desire to invest in the next generation.

**Proposition 3.** A higher parental wage increases the educational inequality between rich and poor children.

**Proof.** As can be seen in Figure 6, the poor and the rich have different attitudes towards increases parental income. The poor parents tend to decrease educational spending and parental time for their children's development, and instead focus more on basic living conditions $\left(\frac{\partial q/z}{\partial w_p}\right)_{poor} < 0$. On the contrary, it is another story for the rich, whose marginal utility is relatively small for investing in themselves, so they choose to invest in their children's education $\left(\frac{\partial q/z}{\partial w_p}\right)_{rich} > 0$. Hence, educational opportunity is balanced.

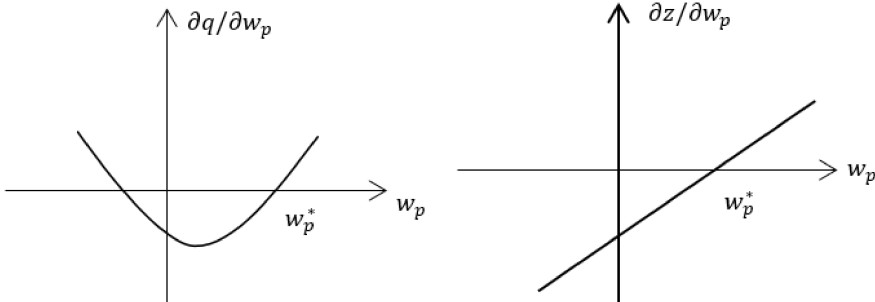

**Figure 6.** Function of household inputs to parental wage rate.

### 2.2.4. Child's Wage Rate

A higher return to education is a good sign for society that the children of less-educated parents could change their fates through education. In this way, increased $w_k$ is presumed to promote social fairness and build a harmonious society. Although both the rich and the poor increase their spending on education, their children receive different nurturing, affected by their parents' socioeconomic status. The rich families provide better conditions and opportunities for their children's development than the poor families do.

**Proposition 4.** There is a complementary effect between children's wages and private spending and parental effort. However, with the different educational conditions offered by their parents, the educational gap is eventually enhanced if the return to education increases.

**Proof.** With an increased return to education, both the rich and the poor increase their household spending on education $\left(\frac{\partial q/z}{\partial w_k}\right)_{poor/rich} > 0$. However, the educational resources and the environments provided for the children from different families are heterogeneous. Because of a better quality of education, children of rich parents are more likely to succeed than those who grow up in a poor family. Therefore, educational inequality increases with the same input in education as shown in Figure 7.

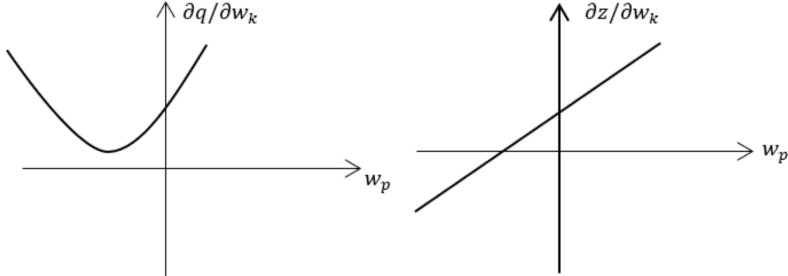

**Figure 7.** Function of household inputs to child's wage rate.

### 2.2.5. Return on Capital

Interest rates are regarded as effective tools to regulate the distribution of household wealth in society. However, poor and rich parents respond differently to increased interest rates, forming the income effect and substitution effect. For the poor parents, the income effect dominates, reflecting that they could spend more on consumption and their children's education. It is the opposite case for the rich, for whom the substitution effect plays a dominant role.

**Proposition 5.** The poor parent increases educational spending when *r* is increased. However, the rich parent decreases their investments in time and money when their child's expected wage is high. Therefore, educational inequality unambiguously decreases.

**Proof.** When a child's expected income is considered low, both rich and poor parents choose to increase household spending on education $\left(\frac{\partial q}{\partial r}\right)_{poor/rich} > 0$ and reduce their parental effort $(\partial z/\partial r)_{poor/rich} < 0$ with increased r (seen from Figure 8), probably because they think accompanying their children is not beneficial, while money is more important for their children's development. On the other hand, when parents are confident about their children's future, there is a crowd-in effect between household spending and capital return for the poor parent $\left(\frac{\partial q/z}{\partial r}\right)_{poor} > 0$, while there is a crowd-out effect for the rich parent $\left(\frac{\partial q/z}{\partial r}\right)_{rich} < 0$. Because of the increasing rate of return, poor adults only need to save less in the first period in order to gain the same income when retired. They have more spare money for consumption and educational investment. The poor parent chooses to spend money on education, q, because it will further improve the next generation's well-being. However, the capital return does not affect the rich at all. Therefore, the educational gap decreases.

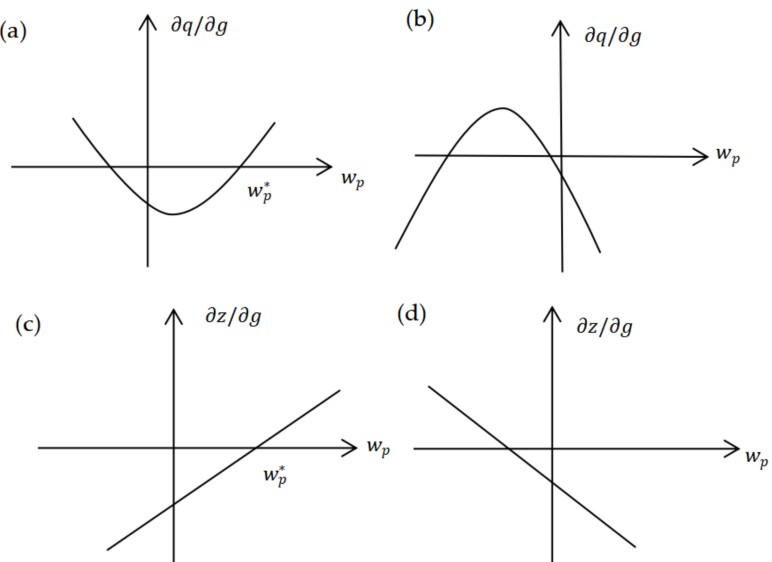

**Figure 8.** Function of household inputs to capital return.

## 3. Data and Empirical Model

### 3.1. Empirical Specification

Regarding household spending, our specification for studying the impact of public education spending on household money allocation is consistent with the equation and uses the following form:

$$M_{ipt} = \alpha_{pt} + \beta G_{kt} + \gamma X_{ipt} + \delta_{ipt} \tag{16}$$

The dependent variable $M_{ipt}$ is the per-student spending on school tuition, textbooks, software, transportation, school choice, food, accommodation, and private tutoring by household, I, living in the province, p, in year, t. $G_{kt}$ is the per-student public education spending on teachers, students, public staff, and infrastructure by the local government, k, in year, t. $\alpha_{pt}$ is the province–year fixed effects, which reflects the time trend of the specific province that will affect both the private and public education spending. $X_{ipt}$ includes the household characteristics that will potentially affect the decision making for education by parents, including household socioeconomic variables, such as the father and mother's education, age, and work, educational saving, average disposable income, and child-level variables, such as the child's gender, age, schooling level, participation in activities, and class ranking.

Regarding parental effort, we estimated the influence of public education spending on household time allocation with the following model:

$$T_{ipt} = \alpha_{pt} + \beta G_{kt} + \gamma X_{ipt} + \delta_{ipt} \tag{17}$$

The dependent variable $T_{ipt}$ is the family care variable by household i living in province p in year t. The specifications for the other variables are the same as before.

We investigated whether private inputs, including time and money, respond differentially to public education spending by parental income level, child gender, region, schooling level, and family size. These dimensions of heterogeneity could reflect the real impact of public education expenditure on family behavior. Section 4 presents the detailed results.

### 3.2. Data

The data we used came from multiple sources. The private education spending and parental time data were extracted from the China Family Panel Studies (CFPS) conducted by the Institute of Social Science Survey (ISSS). In the years 2010, 2012, 2014, and 2016, four rounds of nationwide household surveys were carried out. The target sample of the CFPS consists of 16,000 households in 25 provinces/municipalities/autonomous regions in China (excluding Hong Kong, Macao, Taiwan, Xinjiang, Tibet, Qinghai, Inner Mongolia, Ningxia, and Hainan). All eligible households and household members are subjects of the survey. By collecting data at three levels (individual, family, and community), in which the individual level is classified by age (adult and children), the project aims to document changes in the Chinese society, economy, population, education, and health, so as to provide data for academic research and public policy analyses.

Private education spending is covered in the parental response section of the CFPS children's questionnaire. The data from the years 2010 and 2016 did not have classifications or specific values for each type of private education investment, so we only used the data from the years 2012 and 2014. Household education expenses mainly include nanny fees, child care expenses, school tuition, textbooks, education software fees, transportation fees, school choice fees, food expenses, accommodation fees, and private tutoring. Because nanny fees and child care expenses are not applicable to each stage of education, we only examined the last eight types of the listed family education expenses, of which the first seven items are defined as school education expenditure, and the latter is defined as private tutoring expenditure.

The public education expenditure data came from the China Educational Finance Statistical Yearbook issued by the Ministry of Education and the National Bureau of Statistics. The yearbook divides the total expenditure into business expenditures and capital construction expenditures. The former is divided into personal and public components, in which the personal part includes expenditure on teachers (wages and welfare) and subsidies for children, and the public part includes expenditure on goods and services and other capital expenditures. Based on this category, we divided public education expenditure into four categories—namely, teacher expenditure, student subsidies, public spending, and infrastructure expenditure—in order to analyze the impact of each type of public expenditure on private education investments. We focused on children from ages 0 to 18 years old, which includes four stages of basic education—kindergarten, primary school (Grades 1–6), middle school (Grades 7–9), and high school (Grades 10–12). For different characteristics in each stage, we collected four types of education expenditures in four stages, as well as the number of students at all levels from the National Bureau of Statistics, calculating the per-student public education spending by the local government.

In this study, we mainly restricted the sample to children between 0 and 18 years old, which included the four stages of education—kindergarten, elementary school, junior high school, and high school. This provided us with 6512 observations, covering 26 provinces across the country.

Table 1 reports the summary statistics of per-student education spending by households and governments for each year and educational stage. All monetary values were



comparable in 2012. Due to data limitations, we did not have high school samples from 2012, and by 2014, some middle school students reached high school. For private education spending, from a horizontal perspective, the total educational expenditures and each type increased with the academic qualifications. Spending in middle and high school was much higher than in primary school and kindergarten. From the vertical perspective, from 2012 to 2014, the total expenditures of kindergartens and primary schools increased significantly (77.15 and 28.53%), while there were no apparent changes in the middle schools. Among the total spending, school tuition, textbooks, transportation fees, food expenses, and private tutoring slightly increased at each stage, while education software fees, school choice fees, and accommodation fees decreased slightly. This shows that with increases in income levels and improvements to the education system, families spent more money on their children's education and living expenses and reduced their costs for accommodations and choosing schools.

**Table 1.** Means and standard deviations of private and public education spending.

| | Kindergarten | | Primary School | | Middle School | | High School | |
| | **2012** | **2014** | **2012** | **2014** | **2012** | **2014** | **2012** | **2014** |
|---|---|---|---|---|---|---|---|---|
| *Private education spending* | | | | | | | | |
| Total spending | 1514.34 (3675.40) | 2683.03 (4644.63) | 1923.14 (3111.43) | 2471.76 (4457.41) | 4089.39 (6323.34) | 4348.98 (5789.32) | NA | 6923.55 (6593.45) |
| School tuition | 381 (1100.49) | 414.23 (1266.96) | 610.40 (1381.94) | 715.13 (1653.31) | 1210.11 (3196.95) | 1242.13 (2474.00) | NA | 2357.54 (3322.37) |
| Textbooks | 69.69 (252.47) | 86.19 (276.39) | 214.23 (401.45) | 240.40 (550.40) | 338.04 (391.14) | 362.89 (742.08) | NA | 678.12 (1363.16) |
| Software | 18.66 (165.43) | 16.10 (162.80) | 28.69 (218.91) | 42.18 (383.02) | 60.78 (388.84) | 46.58 (372.35) | NA | 133.80 (530.75) |
| Transportation | 50.12 (269.00) | 79.80 (366.51) | 130.94 (442.31) | 158.50 (504.12) | 197.66 (385.41) | 220.15 (481.65) | NA | 391.38 (771.66) |
| School choice | 22.09 (832.52) | 13.72 (332.50) | 31.66 (419.13) | 29.01 (344.04) | 209.48 (1715.62) | 123.69 (1181.93) | NA | 500.13 (2616.29) |
| Food | 179.66 (878.74) | 272.22 (710.37) | 354.18 (774.36) | 432.83 (909.69) | 1054.89 (1462.97) | 1266.24 (1511.54) | NA | 2211.11 (2272.75) |
| Accommodation | 6.61 (83.33) | 3.87 (67.53) | 34.47 (259.78) | 44.86 (331.77) | 146.80 (1034.39) | 94.95 (483.95) | NA | 230.74 (651.13) |
| Private tutoring | 70.89 (587.39) | 164.41 (1221.16) | 405.46 (1505.60) | 687.16 (2866.74) | 1003.17 (2778.40) | 1010.98 (3424.06) | NA | 1304.35 (2657.74) |
| *Public education spending* | | | | | | | | |
| Total spending | 2586.67 (2116.80) | 4070.45 (2830.27) | 5853.19 (2617.37) | 8110.71 (3026.33) | 9143.13 (4849.07) | 11,542.95 (5425.11) | NA | 15,322.10 (12,198.55) |
| Teacher expenditure | 1348.61 (1169.33) | 1852.40 (1501.36) | 2775.56 (1172.21) | 3506.26 (1278.77) | 4179.79 (2405.72) | 4911.47 (2250.89) | NA | 6099.95 (4959.61) |
| Student subsidy | 144.34 (157.71) | 227.42 (218.57) | 1127.68 (515.53) | 1737.99 (709.77) | 1387.87 (744.12) | 2007.20 (891.59) | NA | 1892.11 (1329.02) |
| Public spending | 1046.81 (806.01) | 1924.31 (1205.19) | 1858.00 (1097.32) | 2744.54 (1350.24) | 3353.45 (1741.96) | 4318.12 (2342.41) | NA | 6986.09 (5904.59) |
| Infrastructure | 46.90 (47.97) | 66.33 (67.46) | 91.96 (68.11) | 121.91 (74.69) | 222.02 (258.74) | 306.15 (256.57) | NA | 343.95 (406.01) |

Notes: Per-student public education at every school level equals total spending divided by the number of students in kindergarten, primary, middle, and high schools, respectively. Numbers in parentheses are standard deviations. The sample included households with students at the basic education level (Grades 1–12). All spending variables are shown in constant 2012 CNY.

Public education investments followed the same trend as private education expenditure, which increased with the academic levels. In 2014, the total education expenditure in high school was CNY 15,322.1, which was about quadruple (CNY 4070.45) that in kindergarten, twice (CNY 8110.71) as much as that in the primary school, and 1.3 times as (CNY 11,542.95) in the middle school. Teacher expenditure and public spending consistently comprised the majority of total expenditure. Student subsidies increased significantly, from

CNY 144.34 and CNY 227.42 in kindergarten to CNY 1127.99 and CNY 1387.87 in primary schools. In each year, the governments spent less on infrastructure expenditure, with only CNY 343.95 for high schools in 2014.

Figures 9 and 10 depict the proportions of private education expenditures and public education spending for all school levels. In kindergartens, household spending consisted mainly of school tuition and textbooks, and school tuition increased by 70% in 2014. The proportions of expenditures in the primary schools did not change much between 2012 and 2014; school tuition and textbooks were the main expenses, and food expenses and private tutoring were the lower expenses. The spending in the middle schools was similar to that in the high schools, and the largest proportions were for school tuition and food expenses, while textbook expenses were relatively low compared with the other stages, and the proportion of private tutoring was greatly increased. With the improvement of children's education, family education expenditures were also more diversified, mainly comprising school tuition and textbooks, and including a variety of expenditures.

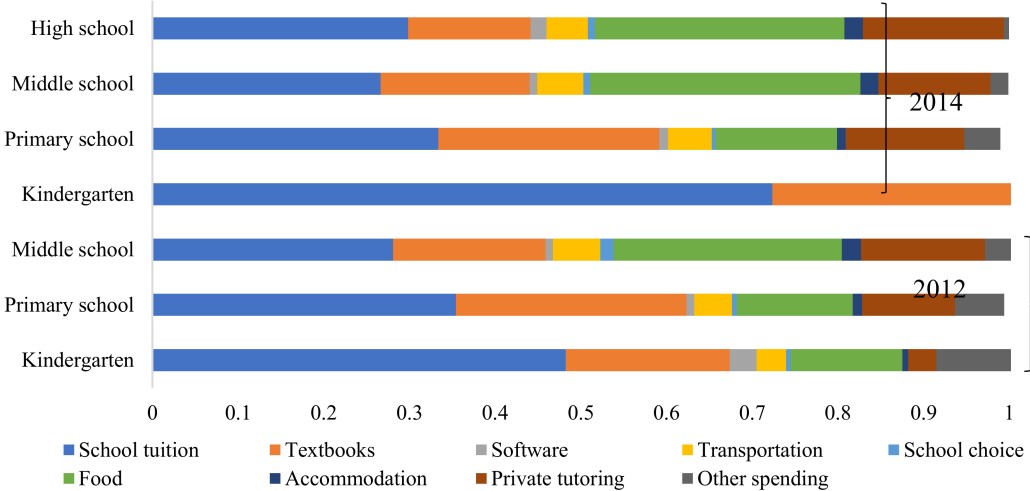

**Figure 9.** Composition of private education spending. The numbers shown above are the proportions of each category spent by households at every school level and year. Household spending variables were calculated for the sample of households with only students at a basic educational level (Grades 1–12). All spending variables are shown in constant 2012 CNY.

Figure 10 shows the composition of public education spending. From a horizontal perspective, teacher expenditure and public spending were major expenditures in every period and degree. Governments spent less on student subsidies in kindergarten and placed the most emphasis on them in primary school. High schools and middle schools spent more on infrastructure than primary schools and kindergartens, though their spending still made up a relatively small amount of total spending. From a vertical perspective, the proportion of each stage did not change significantly. Teacher expenditure declined slightly in each academic stage. Relatively speaking, public expenditure increased slightly.

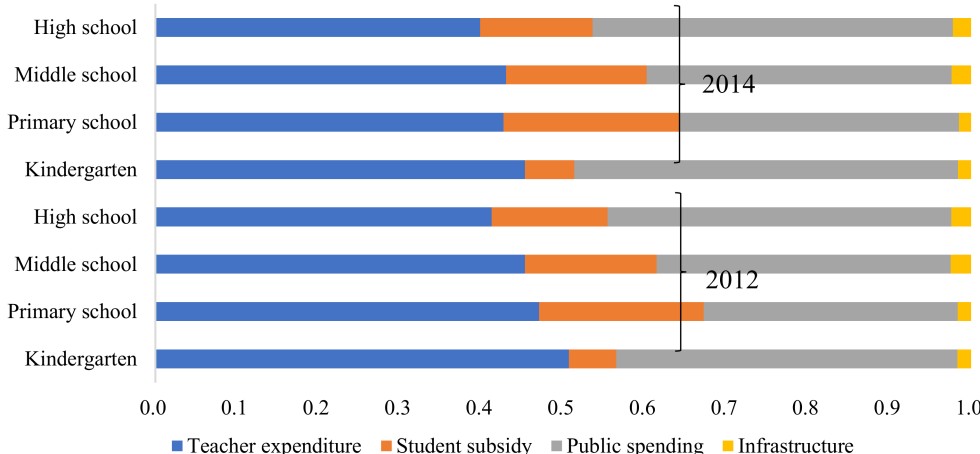

**Figure 10.** Composition of public education spending. The numbers shown above are the proportions of each category spent by the government at every school level and year. Per-student public education at every school level equals total spending divided by the number of students in kindergartens, primary, middle, and high schools, respectively (Grade 1–12). All spending variables are shown in constant 2012 CNY.

Table 2 provides the sample summary statistics of the main variables, with Columns 1 to 3 representing the full sample and the subsamples in the two areas (country and city), respectively. The table shows that cities spent more money on public education spending than rural areas. On average, local urban governments spent CNY 3039.97 and CNY 2533 on teachers and public affairs, respectively, almost 1.3 times higher than the spending of the rural governments. Both rural and urban areas believe that infrastructure is not important for education, and spent CNY 114.58 and CNY 99.13, respectively. In terms of private education spending, that of urban areas was also significantly higher than that of rural areas, except for accommodation fees. The likely cause for this is that children in cities are mainly studying at the local primary school, so they do not need to pay higher accommodation fees. On the contrary, most students in rural areas choose to study in the nearby cities, which costs them extra living expenses. In general, families in cities spent more on school tuition, food expenses, and private tutoring (CNY 785.58, CNY 509.53, and CNY 887.03, respectively). In the countryside, households mainly spent money on school tuition and food expenses (CNY 512.93 and CNY 368.93); however, the expenditure on tutoring was relatively small.

We explored four variables that reflect parental effort—how frequently parents (1) turn down the TV in order to not disturb their child's study, (2) discuss school activities with their child, (3) check their child's homework, and (4) select a TV program for their child. All these variables take on five possible values: 1 indicates "never", 2 indicates "once a month", 3 indicates "once a week", 4 indicates "two to four times per week", and 5 indicates "five to seven times per week". It can be observed in Table 2 that urban households spent more time on children's education than rural households.

In our econometric analysis, the control variables included the socioeconomic characteristics and the respondent's demographics, including their gender (male or female), age in years (age), educational attainment (primary school or below, middle school, high school, and college or above), residential status (urban or rural), employment type (working or unemployed), number of children (family size), and annual personal income (income). Among other control variables, the parents in urban areas had higher education levels and net income per capita than those in rural areas, and were more willing to save money for education. At the same time, the children in urban areas participated in community organizations and served as class leaders. The number of times the children saw their parents was also higher. The urban areas in the data were mainly concentrated in the east, while the rural areas were more concentrated in the west.

**Table 2.** Sample summary statistics for key variables.

| Variable | Definition | Full Sample (1) | City (2) | Country (3) |
|---|---|---|---|---|
| *Explained variable* | | | | |
| Tuition | Annual tuition expenditure | 617.28 (1611.84) | 785.58 (2073.42) | 512.93 (1228.79) |
| Textbook | Annual textbook expenditure | 176.43 (449.49) | 254.11 (607.92) | 127.57 (296.86) |
| Software | Annual software expenditure | 29.69 (268.46) | 56.82 (392.71) | 13.39 (148.35) |
| Transportation | Annual transportation fees | 117.46 (414.77) | 138.39 (488.29) | 104.76 (363.01) |
| School choice | Annual school choosing fee | 41.30 (716.65) | 75.20 (1065.55) | 20.38 (366.84) |
| Food | Annual board expenses | 421.44 (1001.51) | 509.53 (1159.37) | 368.93 (890.08) |
| Accommodation | Annual accommodation cost | 33.16 (311.77) | 27.32 (377.75) | 36.96 (264.94) |
| Private tutoring | Annual private tutoring cost | 400.52 (1978.41) | 887.03 (3063.34) | 102.40 (614.25) |
| Keep quiet | How often have you turned down the TV in order to not disturb your child's study? | 3.69 (1.17) | 3.87 (1.13) | 3.59 (0.47) |
| Discuss activities | How often have you discussed school activities with child? | 3.25 (1.16) | 3.42 (1.10) | 3.14 (1.18) |
| Check homework | How often have you checked your child's homework? | 3.25 (1.32) | 3.43 (1.29) | 3.13 (1.31) |
| Program selection | How often have you selected TV programs for your child? | 2.59 (1.36) | 2.64 (1.36) | 2.58 (1.36) |
| *Core explanatory variable* | | | | |
| Teacher | Annual teacher expenditure | 2623.10 (1879.54) | 3039.97 (2325.59) | 2348.56 (1445.91) |
| Student | Annual student subsidy | 909.30 (888.85) | 1013.32 (994.14) | 842.10 (802.99) |
| Public | Annual public spending | 2137.97 (1708.51) | 2533.60 (2202.55) | 1874.93 (1173.78) |
| Infrastructure | Annual infrastructure cost | 105.53 (134.32) | 114.58 (173.55) | 99.13 (96.62) |
| *Household socioeconomic variables* | | | | |
| Father's education | Number of years the father/male guardian spent in school. | 1.95 (0.89) | 2.28 (0.97) | 1.74 (0.77) |
| Father's work | Working (1 = yes) | 0.85 (0.36) | 0.87 (0.34) | 0.84 (0.37) |
| Mother's education | Number of years the mother/female guardian spent in school. | 1.80 (0.89) | 2.20 (0.99) | 1.55 (0.70) |
| Mother's work | Working (1 = yes) | 0.68 (0.47) | 0.67 (0.47) | 0.69 (0.46) |
| Savings | Saving for education (1 = yes) | 0.2 (0.4) | 0.24 (0.43) | 0.17 (0.38) |
| Income | Per capita annual household net income (CNY) | 11,006.58 (17,519.72) | 14,589.92 (24,987.56) | 8792.82 (10,167.57) |

**Table 2.** *Cont.*

| Variable | Definition | Full Sample (1) | City (2) | Country (3) |
|---|---|---|---|---|
| *Other control variables* | | 2.74 | | |
| Child's education | Number of years the child spent in school. | 0.67 (0.71) | 0.72 (0.75) | 0.64 (0.67) |
| Child's age | Child's age in years | 7.42 (4.04) | 7.56 (4.06) | 7.34 (4.01) |
| Mother's age | Mother's age in years | 33.84 (6.54) | 34.33 (6.18) | 33.54 (6.72) |
| Father's age | Father's age in years | 35.81 (6.69) | 36.25 (6.24) | 35.53 (6.91) |
| Female | Child's gender (1 = female) | 0.53 (0.50) | 0.52 (0.50) | 0.53 (0.50) |
| Region | City household (1 = yes) | 0.19 (0.39) | 0.43 (0.49) | 0.04 (0.20) |
| Organization | Community organization (1 = yes) | 0.11 (0.32) | 0.16 (0.36) | 0.09 (0.28) |
| Leader | Class leader (1 = yes) | 0.33 (0.47) | 0.38 (0.49) | 0.30 (0.46) |
| Ranking | Class ranking | 3.73 (1.19) | 3.69 (1.15) | 3.76 (1.21) |
| Number of children | Number of children | 1.82 (0.94) | 1.63 (0.86) | 1.94 (0.98) |
| East | Eastern state (1 = yes) | 0.35 (0.48) | 0.43 (0.50) | 0.30 (0.46) |
| West | Western state (1 = yes) | 0.30 (0.46) | 0.20 (0.40) | 0.36 (0.48) |
| Urban | Urban residents (1 = yes) | 0.38 (0.49) | - | - |
| Observation | Sample size | 13,007 | 7337 | 6498 |

Source: China Educational Finance Statistical Yearbook, 2012 and 2014; China Family Panel Studies, 2012 and 2014; National Bureau of Statistics, 2012 and 2014. Note: The reported statistics are the sample means with the standard deviations in parentheses. All spending variables are shown in constant 2012 CNY.

## 4. Estimation Results

We first present our estimated results of the effect of public education spending on parental effort and private spending. Then, we present the results of our analysis on the specific items of household education spending, taking into consideration the heterogeneous responses by household income, schooling level, child number, child gender, and region. Because each spending category had numbers with zero, we used a Tobit model to perform all the regressions. All the standard errors are robust and control for the province–year fixed effects.

### 4.1. Results for Parental Effort

Columns 1–4 of Table 3 present the Tobit estimated response of the parental effort equations for the four measures. All specifications control for the province–year fixed effects, as well as other household and child level variables.

As shown in Table 3, increases in public spending are associated with significant growth in parental effort. To be specific, there is substantial consistency with student subsidies and infrastructure expenditure in the results across the four measures. When the government increased the student subsidies, both poor and rich parents increased their parental effort and attention. Among the four measures, with increased student aid, parents tended to spend more time discussing school activities with their children (0.218) and more often selected programs that were more suitable and beneficial for their children to watch (0.178). In contrast, when infrastructure investments were made, parents reduced their concentration on their children's education, mainly because children had access to

better teaching resources at school, such as ideal teaching courseware and facilities, and thus, they could develop the ability of autonomous learning. In this way, parents do not need to pay as much attention to their child's education as before. As for the other two spending measures, they had opposite effects on the four types of parental effort.

**Table 3.** Regression results for alternative measures of parental effort.

| | Keep Quiet | | Discuss Activities | | Check Homework | | Program Selection | |
|---|---|---|---|---|---|---|---|---|
| | (1) | (2) | (3) | (4) | (5) | (6) | (7) | (8) |
| Public education spending | 0.009 ** | | 0.015 *** | | 0.015 *** | | 0.005 | |
| | (0.0044) | | (0.0045) | | (0.0048) | | (0.0056) | |
| Teacher | | −0.049 ** | | −0.001 | | 0.074 *** | | 0.041 |
| | | (0.0232) | | (0.0233) | | (0.0250) | | (0.0291) |
| Student | | 0.056 * | | 0.093 *** | | 0.218 *** | | 0.178 *** |
| | | (0.0309) | | (0.0312) | | (0.0335) | | (0.0389) |
| Public | | 0.068 *** | | 0.009 | | 0.110 *** | | −0.092 *** |
| | | (0.0215) | | (0.0214) | | (0.0229) | | (0.0266) |
| Infrastructure | | −0.289 * | | −0.147 | | −0.170 | | −0.080 |
| | | (0.1489) | | (0.1474) | | (0.1582) | | (0.1835) |
| Father's education | −0.023 | 0.021 | 0.010 | 0.010 | 0.042 | 0.053 ** | −0.008 | 0.038 |
| | (0.0236) | (0.0236) | (0.0241) | (0.0241) | (0.0260) | (0.0259) | (0.0302) | (0.0207) |
| Father's work | 0.145 *** | 0.132 ** | 0.022 | 0.001 | 0.067 | 0.037 | 0.055 | 0.039 |
| | (0.0522) | (0.0521) | (0.0506) | (0.0505) | (0.0545) | (0.0541) | (0.0626) | (0.0624) |
| Mother's education | 0.129 *** | 0.123 *** | 0.198 *** | 0.194 *** | 0.190 *** | 0.201 *** | 0.034 | 0.042 |
| | (0.0258) | (0.0259) | (0.0263) | (0.0264) | (0.0284) | (0.0284) | (0.0330) | (0.0330) |
| Mother's work | 0.039 | 0.038 | −0.023 | −0.030 | −0.012 | −0.025 | 0.007 | −0.010 |
| | (0.0413) | (0.0414) | (0.0405) | (0.0406) | (0.0438) | (0.0435) | (0.0502) | (0.0501) |
| Savings | 0.094 ** | 0.101 ** | 0.284 *** | 0.281 *** | 0.356 *** | 0.346 *** | 0.175 *** | 0.167 *** |
| | (0.0401) | (0.0401) | (0.0394) | (0.0394) | (0.0425) | (0.0422) | (0.0488) | (0.0487) |
| Income | 0.011 | 0.008 | −0.018 | 0.017 | −0.038 ** | 0.029 | −0.034 * | 0.027 |
| | (0.0150) | (0.0151) | (0.0147) | (0.0148) | (0.0159) | (0.0159) | (0.0183) | (0.0183) |
| Child's education | 0.100 | 0.115 ** | 0.036 | 0.062 | 0.191 *** | −0.116 ** | −0.060 | 0.001 |
| | (0.0502) | (0.0516) | (0.0489) | (0.0503) | (0.0527) | (0.0539) | (0.0605) | (0.0621) |
| Child's age | −0.042 *** | −0.041 *** | 0.035 *** | −0.037 *** | 0.108 *** | 0.117 *** | −0.008 | −0.016 |
| | (0.0097) | (0.0098) | (0.0097) | (0.0098) | (0.0105) | (0.0105) | (0.0121) | (0.0121) |
| Mother's age | 0.006 | 0.005 | 0.001 | 0.002 | 0.006 | 0.010 | 0.007 | 0.010 |
| | (0.0056) | (0.0056) | (0.0057) | (0.0057) | (0.0062) | (0.0062) | (0.0072) | (0.0072) |
| Father's age | −0.009 | −0.009 | 0.002 | 0.002 | 0.017 *** | 0.019 *** | 0.001 | −0.009 |
| | (0.0054) | (0.0054) | (0.0055) | (0.0055) | (0.0060) | (0.0060) | (0.0070) | (0.0069) |
| Female | −0.037 | −0.035 | −0.030 | −0.030 | 0.063 * | 0.064 * | −0.090 ** | −0.088 ** |
| | (0.0330) | (0.0330) | (0.0340) | (0.0340) | (0.0369) | (0.0366) | (0.0429) | (0.0427) |
| Region | 0.077 | 0.083 | 0.070 | 0.066 | 0.143 ** | 0.125 ** | 0.106 | 0.090 |
| | (0.0543) | (0.0543) | (0.0555) | (0.0555) | (0.0601) | (0.0597) | (0.0697) | (0.0694) |
| Organization | 0.095 | 0.228 *** | 0.086 | 0.124 * | 0.170 * | 0.228 *** | 0.007 | −0.015 |
| | (0.0811) | (0.0790) | (0.0764) | (0.0696) | (0.0876) | (0.0790) | (0.0964) | (0.0834) |
| Leader | −0.123 ** | 0.054 | −0.007 | 0.115 ** | −0.048 | 0.054 | 0.038 | 0.081 |
| | (0.0586) | (0.0582) | (0.0564) | (0.0511) | (0.0648) | (0.0582) | (0.0713) | (0.0612) |
| Ranking | −0.017 | 0.035 | 0.022 | 0.026 | 0.022 | 0.035 | −0.006 | −0.013 |
| | (0.0233) | (0.0231) | (0.0225) | (0.0202) | (0.0258) | (0.0231) | (0.0284) | (0.0243) |
| Number of children | −0.041 ** | −0.037 * | −0.046 ** | −0.042 ** | 3.035 | −0.040 * | 0.028 | 0.029 |
| | (0.0192) | (0.0193) | (0.0196) | (0.0197) | (0.0213) | (0.0212) | (0.0247) | (0.0247) |
| Urban | 0.099 ** | 0.096 ** | 0.025 | 0.021 | 0.074 * | 0.075 * | 0.009 | 0.010 |
| | (0.0393) | (0.0394) | (0.0402) | (0.0402) | (0.0435) | (0.0432) | (0.0505) | (0.0503) |
| Constant | 3.717 *** | 3.838 *** | 3.198 *** | 3.014 *** | 4.807 *** | 4.263 *** | 2.625 *** | 2.162 *** |
| | (0.2002) | (0.1633) | (0.2012) | (0.1658) | (0.2177) | (0.1783) | (0.2518) | (0.2073) |
| Observations | 4701 | 4701 | 4721 | 4721 | 4719 | 4719 | 4715 | 4715 |

Notes: Clustered robust standard errors are in brackets; all columns were estimated with a Tobit model. All columns control for province–year fixed effects. * Statistical significance at a 10% level. ** Statistical significance at a 5% level. *** Statistical significance at a 1% level.

Consistent with the previous literature, the father's and mother's education and family income had positive correlations with all four measures of parental care. More specifically, mothers had more of an impact on their children's development than fathers. However, this effect was rapidly reduced if the mothers were working, because they could not afford the time to take care of their children. In terms of the parents' ages, the mother's age was positively related to parental effort, while the father's age had the opposite effect—perhaps the mother's behavior plays a bigger role than the father's in their children's development.

Regarding the characteristics of the child, there appeared to be an association between highly-educated children and parental time, with one exception—there was a negative association with the frequency of discussing school activities. Daughters and young children received remarkably more parental care, including a higher frequency of keeping quiet, checking homework, and selecting TV programs (all factors except for the school discussion variable). Joining organizations, serving as a class leader, and the class ranking had positive effects on parental effort, which implies that participating in more school activities gives parents the opportunity to communicate with their children. The number of children was negatively associated with parental time (except for program selection), which may show parental effort constraints.

### 4.2. Results for Household Education Spending

Columns 1–6 of Table 4 present the estimates for household spending on school spending, private tutoring, and total household education spending, using the same specifications as those in Table 3. As can be seen in the first line, the estimate of public education spending is positive and significant at the 1% level; with an increase of CNY 100 on public spending, household spending on schooling, private tutoring, and total education increased by CNY 88, CNY 117, and CNY 116, respectively. This finding contrasts with the results of recent research that showed that public spending on education crowds out the total private contributions [20].

As can be seen in Column 2, increases in public spending on student subsidies and infrastructure were negatively associated with school spending. With an increase of CNY 1000 on student subsidies and infrastructure spending, household expenditures on school staff decreased by CNY 382 and CNY 1109. This may be due to government subsidies; some primary and middle schools will reduce or exempt tuition fees, so parents will spend less on their school tuition fees. On the other hand, with increasing public spending on teachers and public staff, parents increased spending by CNY 216 and CNY 166 for schooling. When teachers are paid more, their educational attitude is more rigorous and the quality of teaching is also improved. Because teachers are models for their students, students' academic performance improves with better teaching methods. With these conditions, parents are inclined to invest more in sending their children to a better school to obtain a better education.

Column 4 presents the estimates for household spending on private tutoring using the same specifications as before. With an increase of CNY 1000 on teachers' salaries, the average household reduced tutoring spending by CNY 230, which is significant at the 1% level. This result is in line with our hypothesis. When students receive enough education in class, parents do not burden their children with extra private tutoring. The other three types of public spending had positive effects on household education spending. Among them, parents responded intensely to infrastructure spending by increasing their spending on private tutoring by CNY 2462.

**Table 4.** Regression results for household per-student spending on schooling, private tutoring, and total education spending.

| | Schooling Spending | | Private Tutoring Spending | | Total Education Spending | |
|---|---|---|---|---|---|---|
| | (1) | (2) | (3) | (4) | (5) | (6) |
| Public education spending | 88.383 *** | | 116.989 *** | | 166.093 *** | |
| | (11.2870) | | (16.4785) | | (13.5094) | |
| Teacher | | 216.527 *** | | −230.335 *** | | 87.134 |
| | | (52.9588) | | (82.8833) | | (63.2953) |
| Student | | −381.677 *** | | 635.772 *** | | −189.678 ** |
| | | (73.1040) | | (115.2771) | | (87.2346) |
| Public | | 165.922 *** | | 417.158 *** | | 371.638 *** |
| | | (51.8135) | | (80.6210) | | (61.7606) |
| Infrastructure | | −1108.707 *** | | 2461.608 *** | | −518.835 |
| | | (353.7504) | | (519.9478) | | (420.9591) |
| Father's education | 156.598 *** | 114.799 ** | 301.525 *** | 322.019 *** | 250.489 *** | 207.982 *** |
| | (52.8505) | (52.4549) | (93.9597) | (93.4630) | (63.4603) | (62.8350) |
| Father's work | 24.618 | 43.192 | −185.884 | −175.654 | −26.323 | 2.416 |
| | (108.3379) | (107.0369) | (207.1138) | (205.4686) | (128.8208) | (127.2257) |
| Mother's education | 255.466 *** | 219.743 *** | 730.534 *** | 722.180 *** | 490.904 *** | 443.676 *** |
| | (57.2586) | (56.8039) | (98.2191) | (97.5377) | (68.7510) | (68.0403) |
| Mother's work | 290.622 *** | 309.012 *** | 159.473 | 152.586 | 331.470 *** | 322.900 *** |
| | (85.1355) | (84.4588) | (157.8461) | (157.0713) | (101.3432) | (100.4456) |
| Saving | 449.523 *** | 441.739 *** | 657.274 *** | 736.748 *** | 588.984 *** | 601.634 *** |
| | (86.9747) | (86.0913) | (145.4390) | (144.6180) | (103.3512) | (102.2227) |
| Income | 167.611 *** | 0.020 *** | 412.367 *** | 0.025 *** | 249.002 *** | 0.032 *** |
| | (32.9789) | (0.0025) | (66.0620) | (0.0036) | (39.2254) | (0.0030) |
| Child's education | −81.500 | 82.537 | −397.564 ** | −272.815 | −359.009 *** | −186.239 |
| | (108.9077) | (109.5194) | (186.7520) | (187.0540) | (129.3509) | (130.1202) |
| Child's age | 351.524 *** | 366.277 *** | 452.764 *** | 415.805 *** | 461.849 *** | 473.245 *** |
| | (21.4540) | (21.5636) | (39.7914) | (40.1780) | (25.6535) | (25.7431) |
| Mother's age | 6.564 | 0.273 | 20.710 | 22.191 | −3.825 | −7.657 |
| | (13.0179) | (12.9471) | (24.8804) | (24.7752) | (15.7239) | (15.5888) |
| Father's age | 1.664 | 5.184 | −9.782 | −12.456 | 16.352 | 18.182 |
| | (12.6353) | (12.5049) | (24.1222) | (23.9067) | (15.2558) | (15.0517) |
| Female | −91.695 | −95.643 | −272.581 * | −260.955 * | −145.676 | −145.971 |
| | (76.0871) | (75.3358) | (140.8601) | (139.8682) | (91.8689) | (90.6837) |
| Region | 20.350 | 31.349 | 1555.078 *** | 1618.354 *** | 699.405 *** | 722.107 *** |
| | (122.5301) | (121.2482) | (196.6569) | (195.4604) | (146.8452) | (144.9770) |
| Organization | 150.786 | 563.755 *** | 303.443 *** | 1099.935 *** | 365.272 | 1129.972 *** |
| | (255.9277) | (215.7017) | (355.4641) | (428.7480) | (309.907) | (302.6265) |
| Leader | −141.449 | −48.349 | 725.646 *** | 909.864 *** | 111.675 | 228.247 |
| | (186.1543) | (157.4469) | (281.0086) | (346.0686) | (227.0713) | (224.7838) |
| Ranking | −39.912 | −34.305 | −279.563 ** | −15.500 | −142.258 | −58.378 |
| | (74.3185) | (62.5010) | (115.4946) | (143.3468) | (90.8708) | (90.0240) |
| Number of Children | −144.776 *** | −129.757 *** | −614.051 *** | −611.776 *** | −185.857 *** | −165.778 *** |
| | (42.3045) | (41.8938) | (95.6981) | (95.3209) | (50.9481) | (50.3266) |
| Urban | 249.286 *** | 213.830 ** | 1146.420 *** | 1117.475 *** | 397.134 *** | 355.612 *** |
| | (88.7027) | (87.9053) | (162.7832) | (161.7798) | (106.6876) | (105.4815) |
| Constant | −5080.075 *** | −3792.662 *** | 13,864.830 *** | 10,229.73 *** | −7536.839 *** | 5536.908 *** |
| | (422.0767) | (321.9040) | (865.5867) | (652.0480) | (506.8345) | (386.9818) |
| Observations | 8076 | 8076 | 8458 | 8458 | 8070 | 8070 |

Notes: Clustered robust standard errors are in brackets; all columns were estimated with a Tobit model. All columns control for province–year fixed effects. * Statistical significance at a 10% level. ** Statistical significance at a 5% level. *** Statistical significance at a 1% level.

Column 6 presents the results for per-student total private education spending. The pattern of responses by households is identical to the school spending, because private tutoring also plays a very small part in household education spending. Generally speaking, student subsidies and infrastructure spending show a crowd-out effect, mainly for the reduction in school tuition and better educational facilities. On the other hand, public

spending on teachers and public staff has a crowd-in effect on private spending because parents have more motivation to make their children work harder.

In order to obtain robust results, we subdivided the private schooling spending into seven specific items, shown in Table 5, including tuition, textbook, software, transportation, school choice, food, and accommodation. We arrived at the same conclusion as before, that public spending on student subsidies and infrastructure has a crowd-out effect on most private schooling spending. The factors most affected were tuition fees and food spending, which are the basic needs of schools. In contrast to student and infrastructure spending, increases in teacher's wages and public staff only had a slightly positive influence on private schooling spending.

**Table 5.** Regression results for household per-student spending on specific items.

| | Tuition | Textbook | Software | Transportation | School Choice | Food | Accommodation |
|---|---|---|---|---|---|---|---|
| Teacher | 151.768 *** (45.5638) | 49.310 *** (10.3679) | 15.551 *** (4.7069) | 41.897 (28.7776) | −18.303 (12.8004) | 60.537 (43.4011) | 6.585 (4.8710) |
| Student | −207.674 *** (62.3925) | 35.512 ** (14.2750) | −21.115 *** (6.5711) | 48.018 (39.6104) | −34.930 * (17.8654) | −266.365 *** (60.6615) | −9.263 (6.7973) |
| Public | 22.521 (44.6876) | −1.203 (10.3199) | 7.472 (4.7101) | 59.774 ** (27.1477) | 25.967 ** (12.8106) | 223.153 *** (42.1779) | 2.511 (4.8800) |
| Infrastructure | −841.801 *** (302.2737) | −373.338 *** (70.3126) | −52.228 (32.2162) | −745.143 *** (181.8320) | 231.838 *** (87.6298) | −741.996 *** (280.3423) | 11.505 (33.3442) |
| Father's education | −20.835 (45.3056) | 29.484 *** (10.3075) | −3.962 (4.4378) | 54.355 * (31.4197) | 19.003 (12.0690) | 84.268 * (46.4400) | 10.215 ** (4.5930) |
| Father's work | 17.370 (93.1382) | 12.898 (22.7542) | −5.891 (9.4853) | 82.021 (63.9266) | −34.275 (25.7970) | 103.123 (97.8568) | −7.623 (9.8113) |
| Mother's education | 76.296 (49.0981) | 61.465 *** (11.1162) | 13.124 *** (4.8021) | 133.830 *** (33.7318) | 6.514 (13.0608) | 110.511 ** (50.4245) | −11.748 ** (4.9688) |
| Mother's work | 124.932 * (73.7360) | 52.587 *** (17.6099) | 5.991 (7.4299) | −16.306 (49.6272) | −10.396 (20.2004) | 427.335 *** (76.4514) | −1.587 (7.6810) |
| Saving | 112.375 (74.9389) | 70.887 *** (17.6880) | 20.609 *** (7.8001) | 146.149 *** (47.3147) | 65.478 *** (21.2325) | 276.128 *** (73.7963) | 22.805 *** (8.0776) |
| Income | 0.016 *** (0.0021) | 0.001 ** (0.0005) | 0.001 *** (0.0002) | 0.004 *** (0.0012) | 0.001 (0.0006) | 0.006 *** (0.0021) | 0.000 (0.0002) |
| Constant | −3186.907 *** (280.4578) | −889.093 *** (64.5221) | −10.710 (26.1407) | −2061.903 *** (205.2826) | −89.191 (71.0884) | −3876.384 *** (296.7746) | 15.784 (27.0359) |
| Observations | 8227 | 8225 | 8299 | 8293 | 8311 | 8239 | 8263 |

Notes: Clustered robust standard errors are in brackets; all columns were estimated with a Tobit model. All columns control for province–year fixed effects. * Statistical significance at a 10% level. ** Statistical significance at a 5% level. *** Statistical significance at a 1% level.

### 4.3. Heterogeneous Analysis

In this section, we present our examination into how families' responses regarding private money and time allocation differed by the child's gender, the number of children, living region, income level, and schooling level.

The first four columns of Table 6 present the estimated responses of household inputs to increases in public education at different schooling levels. All control variables from Table 3 are included. The results show that, overall, public education investment had a greater impact on the parental effort of children in primary school, with statistical significance, while the coefficients of influence at the secondary school stage were relatively small. Parents pay special attention to the development of their children when they are young. As children grow older, there are some objective reasons that will parents reduce time with their children, such as creating independent consciousness and living at school. Regarding the impact of private spending, increases in public education spending were associated with significant increases in household schooling and total spending in the high schools. With an increase of CNY 1000 in per-student government investments, household spending increased by CNY 276 and CNY 271, respectively. In addition, the impact on private tutoring spending is the most obvious at the middle school level.

**Table 6.** Heterogeneity—results for household per-student spending by the number of children and schooling level.

| | Kindergarten (1) | Primary School (2) | Middle School (3) | High School (4) | One Child (5) | Two Children (6) | Three or More (7) |
|---|---|---|---|---|---|---|---|
| | | | Keep silence | | | | |
| Public education spending | −0.007 (0.0542) | 0.018 *** (0.0063) | 0.003 (0.0070) | 0.008 (0.0114) | 0.016 *** (0.0051) | 0.008 (0.0086) | −0.017 (0.0178) |
| Observations | 5980 | 5421 | 1525 | 470 | 5691 | 5057 | 2272 |
| | | | Discuss activities | | | | |
| Public education spending | 0.032 (0.0528) | 0.025 *** (0.0064) | 0.005 (0.0067) | −0.003 (0.0094) | 0.023 ** (0.0084) | 0.013 *** (0.0050) | −0.025 ** (0.0169) |
| Observations | 5980 | 5421 | 1525 | 470 | 5691 | 5057 | 2272 |
| | | | Check homework | | | | |
| Public education spending | 0.0767 (0.0536) | 0.033 *** (0.0067) | 0.001 (0.0079) | −0.025 * (0.0144) | 0.012 ** (0.0056) | 0.009 (0.0090) | −0.022 (0.0172) |
| Observations | 5980 | 5421 | 1525 | 470 | 5691 | 5057 | 2272 |
| | | | Program selection | | | | |
| Public education spending | −0.0542 (0.0617) | 0.023 *** (0.0029) | −0.009 (0.0087) | −0.004 (0.0185) | 0.013 (0.0105) | 0.009 (0.0066) | −0.011 (0.0187) |
| Observations | 5980 | 5421 | 1525 | 470 | 5691 | 5057 | 2272 |
| | | | Schooling spending | | | | |
| Public education spending | 56.678 ** (23.8324) | 134.580 *** (14.5980) | 61.025 ** (26.0654) | 275.724 *** (78.7844) | 100.722 *** (16.4294) | 51.380 *** (15.1197) | 39.125 (28.2177) |
| Observations | 5980 | 5421 | 1525 | 470 | 5691 | 5057 | 2272 |
| | | | Private tutoring spending | | | | |
| Public education spending | −41.454 (37.8924) | 173.778 *** (22.2487) | 135.288 *** (34.0795) | 19.177 (86.9459) | 104.031 *** (21.0162) | 62.652 *** (23.9219) | 28.452 *** (17.3498) |
| Observations | 5980 | 5421 | 1525 | 470 | 5691 | 5057 | 2272 |
| | | | Total education spending | | | | |
| Public education spending | 52.205 ** (26.1006) | 245.025 *** (18.0612) | 153.1217 *** (32.3558) | 270.619 *** (96.0149) | 177.536 *** (20.2536) | 89.296 *** (16.5395) | 60.288 ** (29.7289) |
| Observations | 5904 | 5221 | 1425 | 470 | 5691 | 5057 | 2272 |

Notes: Clustered robust standard errors at the province level are in brackets; all columns were estimated with a Tobit model. Control variables are the same as those in Table 3. * Statistical significance at a 10% level. ** Statistical significance at a 5% level. *** Statistical significance at a 1% level.

The last three columns capture the differences between the number of children. The regression coefficients for parental effort from Column 7 are negative, which means that parents paid less attention to each child with an increasing number of children. Private education spending showed the same tendencies with the parental effort—household spending on one child was significantly higher than spending on two or three children.

The first three rows of Table 7 contain the Tobit estimations of household inputs by income tertiles. It is obvious that government spending was positively associated with parental money and time allocation in middle-class and high-income families. When it comes to poor families, the coefficients estimated for these variables are negative and significant at a 1% level in the first line. With increased public spending, households tended to decrease their time and money on their children's education. This could be because the rich can get higher returns from investing in their children, while the poor care more about their welfare, and instead decrease their investments in their children. In conclusion, there is an apparent crowd-in effect for middle-class and high-income families, while a crowd-out effect for low-income families. Therefore, the educational gap will increase between the poor and the rich. This finding appears to be consistent with the results of some other studies. Gioacchino showed that wealthier families tended to oppose

the redistributive effect of public spending by increasing their investments in their children. Moreover, worries about "falling" from the middle and upper classes may entice them to spend more to ensure their children's future [21]. The gap in parental investments matches the findings [22] that the gap in test scores between the top and bottom of the income distribution has grown over time.

Table 7 shows the comparison between public education spending and the gender and region indicators. The results show that with girls, households increased their inputs to children when public education spending was higher, while households with boys chose to decrease their spending. This pattern is consistent with the traditional research results. Perhaps girls grow up faster than boys, tend to be more orderly, and are likely to be better at language, and hence, the marginal benefits of education are higher for girls than for boys [23]. Meanwhile, women now out-rank men in higher education and have parents shifted from a heavier investment in boys to girls. Similarly, the long-standing gap between girls' and boys' mathematical performances provides an explanation [24]. From the angle of regional culture, parents in the city took advantage of the higher returns of their children and increased their inputs.

**Table 7.** Heterogeneity—results for household per-student spending by household income, gender, and region.

| | Parental Time Allocation | | | | Parental Money Allocation | | |
|---|---|---|---|---|---|---|---|
| | Keep Silence | Discuss Activities | Check Homework | Program Selection | SchoolingSpending | Private Tutoring Spending | Total Education Spending |
| | (1) | (2) | (3) | (4) | (5) | (6) | (7) |
| Per student public education spending*inc q1 | −0.012 *** (0.0051) | −0.016 *** (0.0051) | −0.023 *** (0.0056) | −0.009 *** (0.0064) | −74.682 *** (12.9349) | −66.940 *** (19.5622) | −113.158 *** (15.3457) |
| Per student public education spending *inc q2 | 0.005 (0.0053) | 0.013 ** (0.0053) | 0.008 (0.0057) | 0.001 (0.0066) | 78.122 *** (13.4652) | 124.739 *** (18.5610) | 172.507 *** (15.9458) |
| Per student public education spending *inc q3 | 0.011 (0.0081) | 0.021 *** (0.0080) | 0.008 (0.0087) | 0.005 (0.0100) | 171.399 *** (20.1211) | 222.641 *** (26.7688) | 342.241 *** (23.7622) |
| Public education spending | 0.037 (0.0316) | 0.042 (0.0324) | 0.077 ** (0.0338) | 0.131 *** (0.036) | 107.460 ** (49.3249) | 149.699 ** (72.2656) | 117.715 ** (52.1273) |
| Gender (boy = 1) | 0.172 (0.2488) | −0.245 (0.2551) | −0.132 (0.2644) | 0.437 (0.0589) | −332.014 (281.5818) | −960.111 * (509.3865) | −387.761 (297.2591) |
| Public education spending *Gender (boy = 1) | −0.026 (0.0363) | 0.036 (0.0370) | 0.031 (0.0385) | −0.058 (0.0409) | 73.026 (47.2255) | 107.146 (74.8934) | 84.222 * (49.8844) |
| Region (city = 1) | 0.638 ** (0.2609) | −0.223 (0.2616) | 0.307 (0.2701) | 0.763 *** (0.2869) | 732.549 ** (297.6012) | 464.062 (536.6911) | 691.8813 ** (314.0635) |
| Public education spending *Gender (city = 1) | 0.074 ** (0.0350) | −0.022 (0.0348) | 0.089 ** (0.0360) | 0.153 *** (0.0384) | 114.754 * (48.1085) | 168.296 (74.4956) | 132.046 (50.7804) |
| Observations | 4699 | 4719 | 4717 | 4713 | 8074 | 8456 | 8068 |

Notes: Clustered robust standard errors are in brackets; all columns were estimated with a Tobit model. Tertile 1 refers to the lowest income family, and Tertile 3 is the highest income family. Control variables are the same as those in Table 3. * Statistical significance at a 10% level. ** Statistical significance at a 5% level. *** Statistical significance at a 1% level.

## 5. Conclusions

This paper mainly discusses the relationship between public education spending and household inputs from theoretical and empirical perspectives. We developed an educational production function that includes household spending and parental effort as inputs for children's human capital. Our theoretical model provides explanations for how rich and poor parents respond differently to public education spending and other economic factors. Rich families have more incentives to invest in their children, suggesting a crowd-in effect of public resources. In contrast, poor families have a crowd-out effect on public spending, as they care more about their well-being. Moreover, the results show that educational investments in parents have spill-over effects on their children, but the degree of influence is different for the poor and the rich. In the same way, a higher return to education encourages parents to spend more on their children, but children from well-off

families have a large advantage in receiving a better education. Thus, the indifferent public education spending policy may not expectedly promote educational equality between rich and poor children.

From the empirical results, we found that there is a significant complementary effect between household inputs and public investments of both time and money in the educational process. Meanwhile, the middle and upper classes prefer higher levels of public spending on education, which echoes the theoretical analysis. Considering the heterogeneity of the whole sample, families with only one child are likely to spend more on girls in the city.

Our work only completes the first step in explaining the responses of parents with different income levels and parental human capital to public education spending. In future work, we will investigate the relationship between public spending and education outcomes at the micro-level. In addition, we will also examine how public and private education spending affects the economy's long-term outcomes, and try to find the optimal allocators of resources.

**Author Contributions:** Conceptualization, S.Y. and X.Z.; methodology, S.Y.; software, S.Y.; validation, S.Y.; formal analysis, S.Y.; investigation, S.Y.; resources, X.Z.; data curation, S.Y.; writing—original draft preparation, S.Y.; writing—review and editing, X.Z.; visualization, S.Y.; supervision, X.Z.; project administration, X.Z.; funding acquisition, X.Z. All authors have read and agreed to the published version of the manuscript.

**Funding:** This research was funded by Beijing Municipal Social Science Foundation, grant number 17JDYJA021 and The Social Science Foundation of Chinese Ministry of Education was funded by 18JZD029.

**Institutional Review Board Statement:** The study was conducted according to the guidelines of the Declaration of School of Economics Peking University. All participants, were previously informed.

**Informed Consent Statement:** Informed consent was obtained from all subjects involved in the study.

**Data Availability Statement:** The data are not publicly available due to ethical reasons.

**Acknowledgments:** The authors would like to thank Yinhe Liang from the Central University of Finance and Economics, China, for his support in the research carried out.

**Conflicts of Interest:** The authors declare no conflict of interest.

## Appendix A. The Interior Solution

The following is the interior solution, with zero bequests:

$$
\begin{bmatrix}
u_1'' + \rho u_k'' w_k^2 f_q^2 + \rho u_k' w_k f_{qq} & u_1'' w_p h_p + \rho u_k'' w_k^2 f_q f_z + \rho u_k' w_k f_{qz} & u_1'' \\
u_1'' w_p h_p + \rho u_k'' w_k^2 f_z f_q + \rho u_k' w_k f_{zq} & u_1'' w_p^2 h_p^2 + \rho u_k'' w_k^2 f_z^2 + \rho u_k' w_k f_{zz} & u_1'' w_p h_p \\
u_1'' & u_1'' w_p h_p & u_1'' + u_2'' (1+r)^2
\end{bmatrix}
\begin{bmatrix} dq \\ dz \\ ds \end{bmatrix}
$$

$$
= - \begin{bmatrix}
\rho u_k'' w_k^2 f_q f_g + \rho u_k' w_k f_{qg} & -u_1'' w_p (1-z) + \rho u_k'' w_k^2 f_q f_{h_p} + \rho u_k' w_k f_{qh_p} & -u_1'' h_p (1-z) & \rho u_k'' w_k f_q h_k + \rho u_k' f_q & 0 \\
\rho u_k'' w_k^2 f_z f_g + \rho u_k' w_k f_{zg} & -u_1' w_p - u_1'' w_p^2 (1-z) h_p + \rho u_k'' w_k^2 f_z f_{h_p} + \rho u_k' w_k f_{zh_p} & -u_1' h_p - u_2'' h_p^2 (1-z) w_p & \rho u_k'' w_k f_z h_k + \rho u_k' f_z & 0 \\
0 & -u_1'' w_p (1-z) & -u_1'' h_p (1-z) & 0 & u_2'' (1+r) s
\end{bmatrix}
\begin{bmatrix} dg \\ dh_p \\ dw_p \\ dw_k \\ dr \end{bmatrix} \tag{A1}
$$

The determinant of the matrix on the left-hand side is

$$
\left| B_1 \right| = \alpha w_p^2 + \beta w_p + \varepsilon < 0 \tag{A2}
$$

where $\alpha < 0$, $\beta < 0$, $\varepsilon < 0$.

***Public education g***

$$
\frac{\partial q}{\partial g} = \frac{-(u_1'' u_2'' w_p^2 h_p^2 (1+r)^2 + (\rho u_k'' w_k^2 f_z^2 + \rho u_k' w_k f_{zz})(u_1'' + u_2'' (1+r)^2))(\rho u_k'' w_k^2 f_q f_g + \rho u_k' w_k f_{qg})}{|B_1|} + \frac{-(u_1'' u_2'' w_p h_p (1+r)^2 + (\rho u_k'' w_k^2 f_z f_q + \rho u_k' w_k f_{zq})(u_1'' + u_2'' (1+r)^2))(\rho u_k'' w_k^2 f_z f_g + \rho u_k' w_k f_{zg})}{|B_1|} \tag{A3}
$$

$$\frac{\partial z}{\partial g} = \frac{-(u_1'' u_2'' w_p h_p (1+r)^2 + (\rho u_k'' w_k^2 f_q f_z + \rho u_k' w_k f_{qz})(u_1'' + u_2''(1+r)^2))(\rho u_k'' w_k^2 f_q f_g + \rho u_k' w_k f_{qg})}{|B_1|} + $$
$$\frac{-(u_1'' u_2''(1+r)^2 + (\rho \rho u_k'' w_k^2 f_q^2 + \rho u_k' w_k f_{qq})(u_1'' + u_2''(1+r)^2))(\rho u_k'' w_k^2 f_z f_g + \rho u_k' w_k f_{zg})}{|B_1|} \tag{A4}$$

### Parental human capital $h_p$

$$\frac{\partial q}{\partial h_p} = \frac{-(u_1'' u_2'' w_p^2 h_p^2 (1+r)^2 + (\rho u_k'' w_k^2 f_z^2 + \rho u_k' w_k f_{zz})(u_1'' + u_2''(1+r)^2))(-u_1'' w_p(1-z) + \rho u_k'' w_k^2 f_q f_{h_p} + \rho u_k' w_k f_{qh_p})}{|B_1|} + $$
$$\frac{-(u_1'' u_2'' w_p h_p(1+r)^2 + (\rho u_k'' w_k^2 f_z f_q + \rho u_k' w_k f_{zq})(u_1'' + u_2''(1+r)^2))(-u_1' w_p - u_1'' w_p^2(1-z)h_p + \rho u_k'' w_k^2 f_z f_{h_p} + \rho u_k' w_k f_{zh_p})}{|B_1|} + \frac{-((u_k'' w_k^2 f_z f_q + u_k' w_k f_{zq})w_p h_p - (u_k'' w_k^2 f_z^2 + u_k' w_k f_{zz}))(-\rho u_1''^2 w_p(1-z))}{|B_1|} \tag{A5}$$

$$\frac{\partial z}{\partial h_p} = \frac{-(u_1'' u_2'' w_p h_p(1+r)^2 + (\rho u_k'' w_k^2 f_q f_z + \rho u_k' w_k f_{qz})(u_1'' + u_2''(1+r)^2))(-u_1'' w_p(1-z) + \rho u_k'' w_k^2 f_q f_{h_p} + \rho u_k' w_k f_{qh_p})}{|B_1|} + $$
$$\frac{-(u_1'' u_2''(1+r)^2 + (\rho u_k'' w_k^2 f_q^2 + \rho u_k' w_k f_{qq})(u_1'' + u_2''(1+r)^2))(-u_1' w_p - u_1'' w_p^2(1-z)h_p + \rho u_k'' w_k^2 f_z f_{h_p} + \rho u_k' w_k f_{zh_p})}{|B_1|} + \frac{-((u_k'' w_k^2 f_q^2 + u_k' w_k f_{qq})w_p h_p - (u_k'' w_k^2 f_q f_z + u_k' w_k f_{qz}))(-\rho u_1''^2 w_p(1-z))}{|B_1|} \tag{A6}$$

### Parental wage rate $w_p$

$$\frac{\partial q}{\partial w_p} = \frac{-(u_1'' u_2'' w_p^2 h_p^2 (1+r)^2 + (\rho u_k'' w_k^2 f_z^2 + \rho u_k' w_k f_{zz})(u_1'' + u_2''(1+r)^2))(-u_1'' h_p(1-z))}{|B_1|} + $$
$$\frac{-(u_1'' u_2'' w_p h_p(1+r)^2 + (\rho u_k'' w_k^2 f_z f_q + \rho u_k' w_k f_{zq})(u_1'' + u_2''(1+r)^2))(-u_1' h_p - u_1'' h_p^2(1-z)w_p)}{|B_2|} + \frac{-((u_k'' w_k^2 f_z f_q + u_k' w_k f_{zq})w_p h_p - (u_k'' w_k^2 f_z^2 + u_k' w_k f_{zz}))(-\rho u_1''^2 h_p(1-z))}{|B_2|} \tag{A7}$$

$$\frac{\partial z}{\partial w_p} = \frac{-(u_1'' u_2'' w_p h_p(1+r)^2 + (\rho u_k'' w_k^2 f_q f_z + \rho u_k' w_k f_{qz})(u_1'' + u_2''(1+r)^2))(-u_1'' h_p(1-z))}{|B_1|} + $$
$$\frac{-(u_1'' u_2''(1+r)^2 + (\rho u_k'' w_k^2 f_q^2 + \rho u_k' w_k f_{qq})(u_1'' + u_2''(1+r)^2))(-u_1' h_p - u_1'' h_p^2(1-z)w_p)}{|B_1|} + \frac{-((u_k'' w_k^2 f_q^2 + u_k' w_k f_{qq})w_p h_p - (u_k'' w_k^2 f_q f_z + u_k' w_k f_{qz}))(-\rho u_1''^2 h_p(1-z))}{|B_1|} \tag{A8}$$

### Child's wage rate $w_k$

$$\frac{\partial q}{\partial w_k} = \frac{-(u_1'' u_2'' w_p^2 h_p^2 (1+r)^2 + (\rho u_k'' w_k^2 f_z^2 + \rho u_k' w_k f_{zz})(u_1'' + u_2''(1+r)^2))(\rho u_k'' w_k f_q h_k + \rho u_k' f_q)}{|B_1|} + $$
$$\frac{-(u_1'' u_2'' w_p h_p(1+r)^2 + (\rho u_k'' w_k^2 f_z f_q + \rho u_k' w_k f_{zq})(u_1'' + u_2''(1+r)^2))(\rho u_k'' w_k f_z h_k + \rho u_k' f_z)}{|B_1|} \tag{A9}$$

$$\frac{\partial z}{\partial w_k} = \frac{-(u_1'' u_2'' w_p h_p(1+r)^2 + (\rho u_k'' w_k^2 f_q f_z + \rho u_k' w_k f_{qz})(u_1'' + u_2''(1+r)^2))(\rho u_k'' w_k f_q h_k + \rho u_k' f_q)}{|B_1|} + $$
$$\frac{-(u_1'' u_2''(1+r)^2 + (\rho u_k'' w_k^2 f_q^2 + \rho u_k' w_k f_{qq})(u_1'' + u_2''(1+r)^2))(\rho u_k'' w_k f_z h_k + \rho u_k' f_z)}{|B_1|} \tag{A10}$$

### Return on capital $r$

$$\frac{\partial q}{\partial r} = \frac{-((u_k'' w_k^2 f_z f_q + u_k' w_k f_{zq})w_p h_p - (u_k'' w_k^2 f_z^2 + u_k' w_k f_{zz}))(\rho u_1'' u_2''(1+r)s)}{|B_1|} \tag{A11}$$

$$\frac{\partial z}{\partial r} = \frac{-((u_k'' w_k^2 f_q^2 + u_k' w_k f_{qq})w_p h_p - (u_k'' w_k^2 f_q f_z + u_k' w_k f_{qz}))(\rho u_1'' u_2''(1+r)s)}{|B_1|} \tag{A12}$$

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
