# Peer review of "How Do Different Households Respond to Public Education Spending?"

_sustainability, doi:10.3390/su132011534_

Round 1

Reviewer 1 Report

The research topic is very interesting. The paper is well-written and comprehensive.

I have two comments:

  1. The authors should get a professional editing service to check the paper before publication.  I come across some typos/errors while reading the paper, which could be fixed easily. for instance, education-al and fam-ily in the abstract. M_ipt in Equation 14, but H_ipt in line 319.  there are many more errors in other places. a comprehensive exam by an outsider is necessary.

2. Why submit this paper to sustainability?  Does the paper have anything related to sustainability?

Author Response

Response 2: Sustainability is an open access journal of economic, and social sustainability of human beings. This paper studies an educational economic problem about how household responses to the increases in public education spending.  Its essence is an intergenerational education transmission.  Many studies suggest that early-life cognitive performance is directly associated with later-life differentiation of income, consumption, and health and even affect the next generation through intergenerational relationships (Heckman, 2006; Heckman et al., 2013; Dettmer et al., 2020). In this view, the impact of public education expenditures on household spending may have a long-term and sustainable influence on student development, which should be taken seriously by the government and society. Therefore, it is consistent with the research purpose of sustainability. 

Reviewer 2 Report

This is a well-conducted study, but the connection with the scope of the journal should be emphasized. 

I would advise authors to seek English language revision.

Author Response

Response 1: Sustainability is an open access journal of economic, and social sustainability of human beings. This paper studies an educational economic problem about how household responses to the increases in public education spending.  Its essence is an intergenerational education transmission.  Many studies suggest that early-life cognitive performance is directly associated with later-life differentiation of income, consumption, and health and even affect the next generation through intergenerational relationships (Heckman, 2006; Heckman et al., 2013; Dettmer et al., 2020). In this view, the impact of public education expenditures on household spending may have a long-term and sustainable influence on student development, which should be taken seriously by the government and society. Therefore, it is consistent with the research purpose of sustainability. 

Response 2: Thanks for the suggestion. In the revised manuscript, we have modified the paragraph, making it more logical and reliable.
